# Recovery of the maternal skeleton after lactation is impaired by advanced maternal age but not by reduced IGF availability in the mouse

**Monika D. Rogowska[1], Uriel N. V. Pena[1], Nimrat Binning[1], Julian K. Christians**[1,2,3,4]*

**1** Department of Biological Sciences, Simon Fraser University, Burnaby, British Columbia, Canada, **2** Centre for Cell Biology, Development and Disease, Simon Fraser University, Burnaby, Canada, **3** British Columbia Children's Hospital Research Institute, Vancouver, British Columbia, Canada, **4** BC Women's Hospital and Health Centre, Women's Health Research Institute, Vancouver, British Columbia, Canada

* julian_christians@sfu.ca

## Abstract

### Background

Lactation results in substantial maternal bone loss that is recovered following weaning. However, the mechanisms underlying this recovery, and in particular the role of insulin-like growth factor 1 (IGF-I), is not clear. Furthermore, there is little data regarding whether recovery is affected by advanced maternal age.

### Methods

Using micro-computed tomography, we studied bone recovery following lactation in mice at 2, 5 and 7 months of age. We also investigated the effects of reduced IGF-I availability using mice lacking PAPP-A2, a protease of insulin-like growth factor binding protein 5 (IGFBP-5).

### Results

In 2 month old mice, lactation affected femoral trabecular and cortical bone, but only cortical bone showed recovery 3 weeks after weaning. This recovery was not affected by deletion of the *Pappa2* gene. The amount of trabecular bone was reduced in 5 and 7 month old mice, and was not further reduced by lactation. However, the recovery of cortical bone was impaired at 5 and 7 months compared with at 2 months.

### Conclusions

Recovery of the maternal skeleton after lactation is impaired in moderately-aged mice compared with younger mice. Our results may be relevant to the long-term effects of breastfeeding on the maternal skeleton in humans, particularly given the increasing median maternal age at childbearing.

**Data Availability Statement:** All relevant data are within the paper and its S1 File and S1–S14 Figs files.

**Funding:** This study was funded by a Natural Sciences and Engineering Research Council of Canada Discovery Grant (JKC; grant number RGPIN-2016-04047) and an NSERC Undergraduate Student Research Award (NB). The funders had no role in study design, data collection and analysis, decision to publish, or preparation of the manuscript.

**Competing interests:** The authors have declared that no competing interests exist.

## Introduction

Lactation has a profound effect on the maternal skeleton. Bone mineral density (BMD) decreases dramatically, regardless of the use of calcium supplements, declining 1–3% *per month* during lactation [1, 2]. In contrast, the decline in BMD is 1.2% *per year* in the first 5 years of menopause [3]. Remarkably, there appears to be complete recovery of the maternal skeleton after weaning [1], and numerous studies have found no long-term effects of parity or lactation history on menopausal risk of low BMD and/or fractures [4, 5]. However, the effects of maternal age on recovery are less clear. It has been hypothesized that pregnancy prior to the acquisition of peak BMD may impair maximum mineralization achieved, and thus may have long-term effects [6]. Some studies have found support for this hypothesis [7–11], although a few have found ambiguous results [12], no effect [5, 13], or the opposite pattern [14].

In contrast to studies of younger mothers, there has been no study of the long-term effects of lactation at advanced maternal age on skeletal health. However, a shorter-term study did find some evidence that recovery after lactation could be impaired at older maternal age; the increase in bone mineral content from parturition to 2 years postpartum was negatively related to age [15]. Given that some bone loss occurs before menopause [16–19] and that the perimenopausal rate of bone loss is higher than that in the early menopause [3], it might be expected that pregnancies at older ages could impact BMD. Such effects would not have been detected in previous studies that found no effect of lactation on menopausal skeletal health because they would not have included the proportions of older mothers that make up today's population as a result of delayed childbearing [20].

In addition to uncertainty regarding the effects of maternal age, the mechanisms underlying the remarkable recovery of the skeleton post lactation are unknown [2]. Insulin-like growth factor 1 (IGF-I) is a likely candidate given its important roles in bone physiology [21–23]. IGF-I availability is regulated by insulin-like growth factor binding proteins (IGFBPs), among which IGFBP-5 is one of the most abundant in bone [24]. IGFBP-5 influences bone mineral density (BMD) [25–27] by regulating IGF availability as well as through IGF-independent effects [28, 29]. The release of IGF-I from IGFBP-5 is regulated by proteases, including pregnancy-associated pregnancy protein-A2 (PAPP-A2) [30–33]. Loss-of-function mutations in the human *PAPPA2* gene cause short stature and reduced bone density [34, 35], and these conditions are improved by treatment with IGF-I [36–38]. In mice, deletion of *Pappa2* reduces the linear growth of bones [39–41] and affects bone composition and microarchitecture [42, 43].

The goal of the present study was to assess the effects of maternal age and *Pappa2* deletion on the recovery of the maternal skeleton after lactation in a mouse model. We predict that recovery after lactation will decrease at older maternal ages, and that recovery will be impaired by deletion of *Pappa2* as a result of reduced IGF availability. Despite differences between rodents and humans [6], changes in the circulating levels of minerals and many of the key hormones involved in calcium homeostasis during pregnancy and lactation are similar in humans and mice [4].

## Materials and methods

### Mice

All work was carried out in accordance with the guidelines of the Canadian Council on Animal Care and was approved by the SFU University Animal Care Committee (protocol 1188–11). Mice were housed in individually ventilated cages (50 air changes/hour; in pairs for breeding and up to 5 mice per cage otherwise) with Enrich-o'Cobs bedding (Andersons Lab Bedding, Maumee, OH) on a 12:12 hour light:dark cycle, at constant temperature (21 ± 1°C), 50%

humidity, with water and food available *ad libitum* (maintenance diet: 5001; breeding diet: Prolab RMH 3000, LabDiet, St. Louis, MO). Animal health was monitored daily, and apart from 9 mice culled due to dystocia or other ill health, there was no mortality outside of planned euthanasia. Euthanasia was performed by isoflurane anesthesia followed by $CO_2$ inhalation.

Mice homozygous for a constitutive *Pappa2* deletion allele (*Pappa2*[KO/KO]) with a C57BL/6 background were generated as previously described [40, 44]. As controls, we used mice homozygous for a conditional deletion allele (where the *Pappa2* gene is intact, but exon 2 is flanked by LoxP sites [40], i.e., floxed, *Pappa2*[fl/fl]). We have previously shown that postnatal growth does not differ between *Pappa2*[fl/fl] and *Pappa2*[wt/wt] mice [44]. The use of *Pappa2*[fl/fl] as controls enabled a reduction in the number of mice used, since we needed to breed these mice to maintain the colony. Genotypes were confirmed by PCR using ear-clip tissue obtained at weaning, as previously described [44].

Mice were bred for the first time at either 2 months, 5 months or 7 months. Mice breed well at 2 and 5 months, but reproductive performance declines by 7 months (personal observation). Peak BMD is achieved shortly before 19 weeks [45], but trabecular bone peaks at 2 months and declines thereafter [46]. Thus, 2 month mice are young breeders, who have not yet achieved peak BMD, 5 month mice have achieved peak BMD and show some trabecular bone loss, while 7 month mice have more bone loss and are approaching reproductive senescence.

Females were euthanized after 3 weeks of lactation (the normal duration of mouse lactation in the lab), or 3 weeks after weaning. In mice, the recovery of the vertebrae is complete by 4 weeks post-weaning, whereas the recovery of the femur is incomplete after 3–4 weeks [47, 48]. We therefore focused on recovery of the femur 3 weeks after weaning as this skeletal site and time point would allow assessment of variation in recovery. Mice bred at 7 months were very poor breeders; many did not become pregnant, and among those who did, many took a long time to become pregnant and/ or cannibalized their litter. Therefore, at 7 months, we only collected mice 3 weeks after weaning to obtain a sufficient sample size. For mice bred at 2 months and 5 months, we only included females who had given birth to, and lactated for, a single litter. However, since breeding performance was so poor at 7 months, at this age we also included females that cannibalized a first litter, but successfully reared a second litter.

Within 4 days of birth, litter size was adjusted to 7 pups to reduce variability in maternal lactational demands. Bred females were excluded from analyses if they weaned fewer than 5 pups. Where possible, for each breeding female, we also collected a virgin age-matched control (AMC), usually a sibling of a bred mouse. All mice (bred and virgin) were kept in identical conditions, including changes to a breeder diet during breeding.

## Micro-computed tomography

Following sacrifice, mice were stored at -20˚C, and later exposed to dermestid beetles for removal of soft tissue. Femurs were scanned using micro-computed tomography (micro-CT) with an isotropic voxel size of 7.4 μm (Scanco Medical μCT100, Switzerland; 70 kVp, 114 μA, 100 ms integration time). For trabecular bone, the region of interest was proximal to the distal growth plate, 222 μm proximal from where the four sections of the bone appeared to fuse in cross-section. The region of interest for cortical bone was the mid-shaft, immediately distal to the third trochanter (where the cross-section of the bone transitioned from a teardrop shape and became rounder). For both trabecular and cortical bone, 5% of the total length of bone was analysed. Apart from knowledge of bone length, the selection of the region of interest was performed blind to age, genotype and breeding status. Measures of trabecular bone microarchitecture included bone volume fraction (BV/TV, %), trabecular number (Tb.N, mm[-1]), trabecular separation (Tb.Sp, μm), and trabecular thickness (Tb.Th, μm) [49]. Measures of cortical

bone morphology included cortical area fraction (Ct.Ar/Tt.Ar, %), average cortical thickness (Ct.Th, μm), and cortical porosity (Ct.Po, %) [49].

### Serum IGF-I and IGFBP-5

At collection, females were blood sampled by cardiac puncture, and serum was stored at -80˚C. We measured serum IGF-I and IGFBP-5 using the IGF-1 DuoSet ELISA kit (#DY791, R&D Systems) and the IGFBP-5 DuoSet ELISA kit (#DY578, R&D Systems), respectively, following the manufacturer's instructions.

### Statistical analyses

Data were analysed using general linear models (proc GLM, SAS, version 9.4). The specific models and sample sizes are described below.

## Results and discussion

### Skeletal traits affected by lactation

To identify traits that were affected by lactation and that showed recovery after three weeks, we first analysed the effects of breeding (bred mice vs. AMC) and timing (collected at wean vs. three weeks after wean) in *Pappa2*<sup>fl/fl</sup> mice (with intact *Pappa2*) at 2 months of age to facilitate comparison with previous studies [47, 48]. These analyses used general linear models including effects of breeding, timing and the interaction between breeding and timing. An effect of breeding indicated that a trait was affected by lactation, whereas an effect of timing indicated a change with age (since it occurred in AMC as well). A significant interaction between breeding and timing potentially indicated recovery from the effects of lactation, if bred mice were more similar to AMC after three weeks of recovery than at wean. Trabecular bone volume fraction was reduced by both breeding and 3 weeks of aging, while trabecular number declined and trabecular separation increased with 3 weeks of aging, and trabecular thickness was reduced by breeding (Table 1). However, none of these traits showed an interaction between breeding and timing, suggesting no evidence of recovery (Table 1). In contrast, cortical area fraction and cortical thickness were reduced by breeding and increased with 3 weeks of aging, but the interaction between breeding and timing was significant, such that values increased more in bred mice over the 3 weeks following weaning, and were more similar to AMC after three weeks of recovery than at wean (Table 1). Cortical porosity showed a similar but inverse pattern (i.e., increased by breeding, etc., Table 1).

Our finding that both trabecular and cortical bone were affected by lactation, but that cortical bone showed more recovery, is consistent with previous work. In a previous study of C57BL/6J mice, femoral trabecular bone volume fraction, trabecular number and trabecular spacing showed little recovery 3 weeks after weaning [47]. In contrast, while cortical thickness, cortical area fraction and cortical porosity still showed an effect of lactation 3 weeks after weaning, the difference between bred mice and non-lactating controls was smaller after recovery than at wean [47]. Similar results were observed after 4 weeks of recovery in CD-1 mice [48]. In rats, tibial cortical bone showed complete recovery 6 weeks after weaning whereas the recovery of trabecular bone was incomplete [50, 51]. The reduction in cortical bone during lactation is due to both reduced periosteal bone formation [52] and increased endocortical bone resorption [53] with the latter being reversed after weaning [54].

**Table 1. Effects of lactation and recovery after weaning on skeletal traits in 2 month old control mice.**

| | Bred females | | Age-matched controls | | Breeding (bred vs. AMC) | Timing (at wean vs. 3 weeks after) | Breeding*timing interaction |
|---|---|---|---|---|---|---|---|
| | At wean | 3 weeks after wean | At wean | 3 weeks after wean | P | P | P |
| Sample size | 17 | 17 | 16 | 15 | | | |
| Trabecular | | | | | | | |
| Bone volume fraction (%) | 6.4±0.4 | 3.9±0.4 | 7.2±0.4 | 4.8±0.4 | 0.04 | < 0.0001 | 0.85 |
| Trabecular number (mm$^{-1}$) | 3.84 ±0.09 | 3.33±0.09 | 3.79 ±0.09 | 3.25±0.09 | 0.45 | < 0.0001 | 0.89 |
| Trabecular separation (μm) | 260±8 | 302±8 | 262±8 | 310±8 | 0.55 | < 0.0001 | 0.78 |
| Trabecular thickness (μm) | 37.4±0.9 | 36.4±0.9 | 43.4±0.9 | 41.4±1.0 | < 0.0001 | 0.11 | 0.59 |
| Cortical | | | | | | | |
| Cortical area fraction (%) | 36.6±0.6 | 43.4±0.6 | 45.5±0.6 | 46.8±0.6 | < 0.0001 | < 0.0001 | < 0.0001 |
| Cortical thickness (μm) | 147±2 | 182±2 | 185±2 | 193±2 | < 0.0001 | < 0.0001 | < 0.0001 |
| Cortical porosity (%) | 6.3±0.2 | 5.1±0.2 | 5.2±0.2 | 5.1±0.2 | 0.008 | 0.006 | 0.01 |

Values are least squares means ± standard error from a general linear model including effects of breeding (bred vs. age-matched control), timing (at wean vs. 3 weeks after), and the breeding*timing interaction.

### Factors affecting circulating IGF-I and IGFBP-5 levels

We analysed circulating IGF-I and IGFBP-5 levels in 2 month old mice using general linear models including the effects of breeding, timing, and genotype, as well as the interaction between breeding and timing (as above), and the three-way interactions between breeding, timing and genotype (to test whether recovery differed between genotypes). IGF-I levels were significantly lower in *Pappa2*$^{KO/KO}$ mice than *Pappa2*$^{fl/fl}$ mice (P = 0.0001), but were not influenced by any other factor (breeding P = 0.28; timing P = 0.13; breeding* timing P = 0.28; breeding*timing*genotype P = 0.92; S1 Fig). Conversely, IGFBP-5 levels were significantly higher in *Pappa2*$^{KO/KO}$ mice than *Pappa2*$^{fl/fl}$ mice (P < 0.0001), but were not influenced by any other factor (breeding P = 0.21; timing P = 0.73; breeding* timing P = 0.51; breeding*timing*genotype P = 0.34; S2 Fig).

The increase in IGFBP-5 levels as a result of the deletion of *Pappa2*, a gene encoding an IGFBP-5 protease, was expected, and consistent with our previous work [42, 44]. Similarly, this increase in IGFBP-5 would be expected to reduce available IGF-I levels, and such an increase has previously been observed in an independent transgenic mouse carrying a mutation eliminating the proteolytic activity of PAPP-A2 [55]. However, while we found that IGF-I levels were not altered by breeding or recovery, a previous study found IGF-I levels to be significantly elevated 3 weeks after weaning, although there was no difference between lactating and non-lactating mice at wean [47].

### Effects of age and *Pappa2* deletion on the recovery of the skeleton after lactation

To examine the effects of age and *Pappa2* deletion on the recovery of the skeleton after lactation, we performed analyses including both *Pappa2*$^{KO/KO}$ and *Pappa2*$^{fl/fl}$ mice at 2 and 5 months of age; sample sizes are shown in Table 2. We did not include 7 month old mice in this

**Table 2. Sample sizes in analyses of the effects of age and *Pappa2* deletion on the recovery of the skeleton after lactation.**

| Age | Breeding | Timing | Genotype | Sample size |
|---|---|---|---|---|
| 2 months | Bred | At wean | Floxed | 17 |
| | | | Knock-out | 14 |
| | | 3 weeks after | Floxed | 17 |
| | | | Knock-out | 19 |
| | AMC | At wean | Floxed | 16 |
| | | | Knock-out | 14 |
| | | 3 weeks after | Floxed | 15 |
| | | | Knock-out | 18 |
| 5 months | Bred | At wean | Floxed | 13 |
| | | | Knock-out | 8 |
| | | 3 weeks after | Floxed | 13 |
| | | | Knock-out | 4 |
| | AMC | At wean | Floxed | 11 |
| | | | Knock-out | 9 |
| | | 3 weeks after | Floxed | 13 |
| | | | Knock-out | 5 |
| 7 months | Bred | 3 weeks after | Floxed | 3 |
| | | | Knock-out | 4 |
| | AMC | 3 weeks after | Floxed | 5 |
| | | | Knock-out | 10 |

analysis since these were not collected at wean. General linear models included effects of breeding, timing, genotype, age, as well as the interaction between breeding and timing (as above), and three-way interactions between breeding, timing and genotype (to test whether recovery differed between genotypes) and between breeding, timing and age (to test whether recovery differed between ages) (Table 3). Trabecular bone volume fraction was reduced by breeding and by aging (both between wean and 3 weeks afterwards, and between 2 and 5 months) (Table 3; Fig 1). The three-way interaction between breeding, timing and age was significant, whereby trabecular bone volume fraction was reduced by breeding at 2 months, but there was little effect of breeding at 5 months, either at wean or 3 weeks later (Table 3; Fig 1). A potential explanation for this result was that trabecular bone was already greatly reduced at 5 months even in virgin mice, such that it could not provide a substantial amount of calcium for lactation. Trabecular number showed a similar pattern, except that this trait declined between wean and recovery at 2 months but not 5 months (Table 3; S3 Fig). Trabecular spacing showed only increases with aging (both between wean and 3 weeks afterwards, and between 2 and 5 months, Table 3; S4 Fig). Trabecular thickness showed an effect of breeding (reduced in bred mice), age (increased at 5 months) and genotype (higher in *Pappa2*$^{KO/KO}$ mice) (Table 3; S5 Fig).

Cortical area fraction showed significant effects of breeding (lower in bred mice), timing (higher after 3 weeks), age (higher at 5 months), genotype (higher in *Pappa2*$^{KO/KO}$ mice), and a significant interaction between breeding and timing, as described above (Table 3; Fig 2). Moreover, there was a significant three-way interaction between breeding, timing and age, whereby the difference between bred and AMC mice diminished between wean and 3 weeks later (reflecting the breeding by timing interaction), but did so to a greater extent at 2 months, indicating that recovery was impaired at 5 months of age (Fig 2). Cortical thickness showed similar patterns, and although the effects of age and genotype were not significant, the three-way

**Table 3. Effects of age and *Pappa2* genotype on the recovery of skeletal traits in 2 and 5 month old mice.**

| | Breeding (bred vs. AMC) | Timing (at wean vs. 3 weeks after) | Age (2 vs. 5 months) | Genotype (*Pappa2*$^{KO/KO}$ vs. *Pappa2*$^{fl/fl}$) | Breeding*timing interaction | Breeding*timing*age interaction | Breeding*timing*genotype interaction |
|---|---|---|---|---|---|---|---|
| Trabecular | | | | | | | |
| Bone volume fraction (%) | 0.02 | <0.0001 | <0.0001 | 0.82 | 0.50 | 0.0012 | 0.14 |
| Trabecular number (mm$^{-1}$) | 0.34 | <0.0001 | <0.0001 | 0.95 | 0.34 | 0.0032 | 0.33 |
| Trabecular separation (μm) | 0.15 | <0.0001 | <0.0001 | 0.84 | 0.18 | 0.14 | 0.37 |
| Trabecular thickness (μm) | <0.0001 | 0.84 | <0.0001 | 0.03 | 0.20 | 0.74 | 0.11 |
| Cortical | | | | | | | |
| Cortical area fraction (%) | <0.0001 | <0.0001 | <0.0001 | <0.0001 | <0.0001 | <0.0001 | 0.36 |
| Cortical thickness (μm) | <0.0001 | <0.0001 | 0.59 | 0.27 | <0.0001 | <0.0001 | 0.54 |
| Cortical porosity (%) | <0.0001 | <0.0001 | 0.0036 | 0.12 | 0.0002 | 0.0013 | 0.97 |

P-values are from general linear models included effects of breeding, timing, age, genotype, the interaction between breeding and timing, and three-way interactions between breeding, timing and genotype (to test whether recovery differed between genotypes) and between breeding, timing and age (to test whether recovery differed between ages). AMC = age-matched controls.

interaction between breeding, timing and age was significant (Table 3; S6 Fig). Cortical porosity also showed similar patterns (Table 3; S7 Fig).

No trait showed a significant three-way interaction between breeding, timing and genotype (Table 3), suggesting that recovery was not impaired by *Pappa2* deletion. While this result is surprising given the roles of IGF-I and IGFBP-5 in bone physiology, IGF-I availability may not be crucial for recovery after lactation. A previous study found that osteocyte-derived IGF-I is not required for the recovery of bone following dietary calcium deprivation [56]. *Pappa2* deletion has previously been shown to affect bone growth in virgin animals, and in the present study we found that it increased cortical area fraction and trabecular thickness. The increase in cortical area fraction is consistent with previous work [42], and is likely the result of reduced IGF-I availability, given that it was also observed with a knock-in *Pappa2* allele coding for protein that lacked proteolytic activity [55].

Because 7 month old mice were not collected at wean, we also performed analyses including *Pappa2*$^{KO/KO}$ and *Pappa2*$^{fl/fl}$ mice at 2, 5 and 7 months of age, including only those mice collected 3 weeks after weaning. General linear models included effects of breeding, age and genotype, as well as the interactions between breeding and age and between genotype and age. Because these analyses included only mice collected 3 weeks after weaning, the effect of breeding was used to assess recovery (a significant effect of breeding 3 weeks after weaning indicated that recovery was not complete), and the breeding by age interaction was used to test whether recovery differed between ages. Similar to the previous analysis, for trabecular bone volume fraction, the effect of age was significant and there was a significant interaction between age and breeding. Trabecular bone volume fraction was reduced by breeding at 2 months, but not at 5 and 7 months (Table 4; Fig 3). For trabecular number and spacing, only the effect of age was significant, with number decreasing and spacing increasing with age (Table 4; S8 and S9

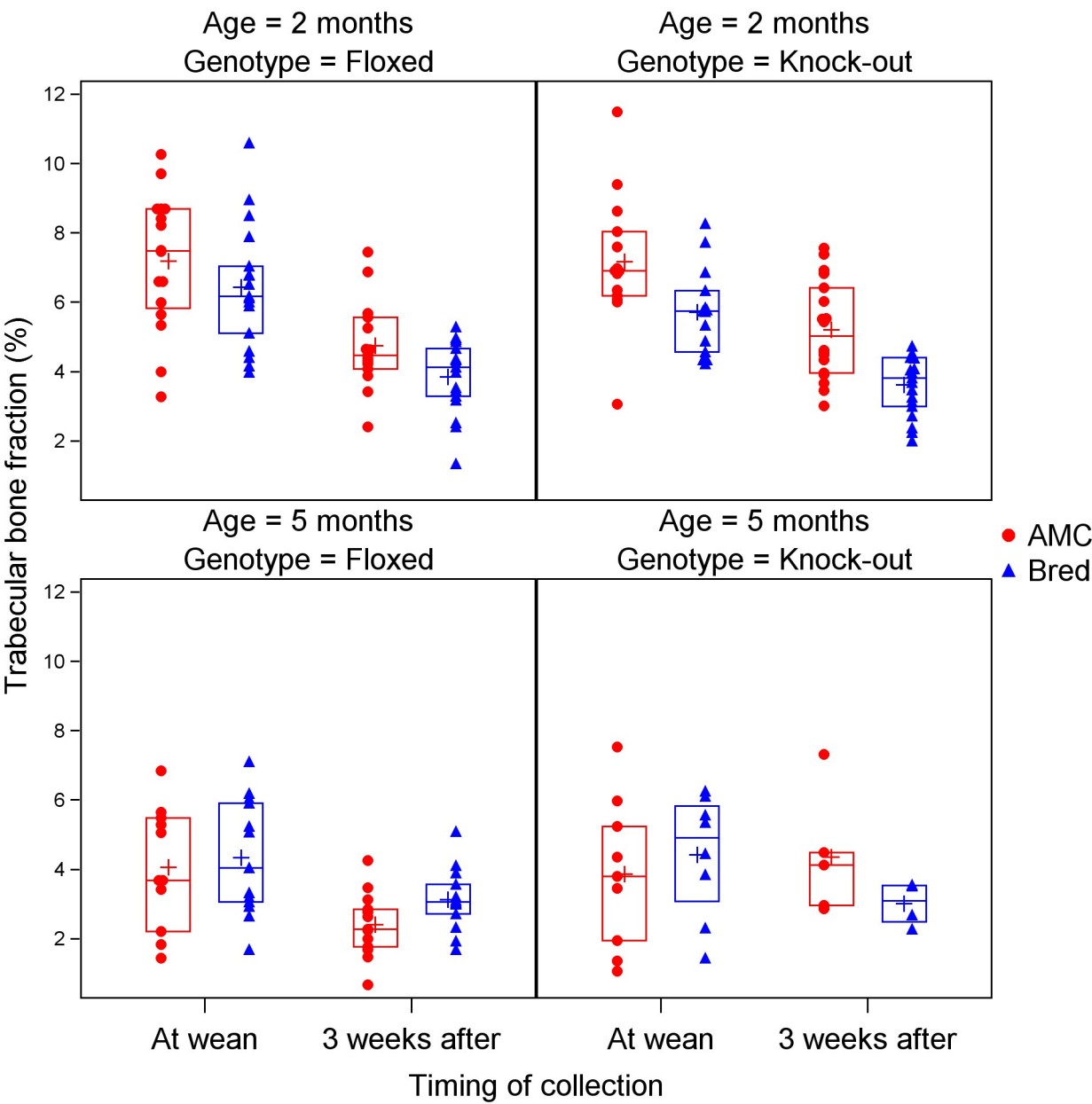

**Fig 1. Effects of age, *Pappa2* genotype, lactation and recovery after weaning on trabecular bone fraction.** Blue triangles denote bred mice, and red circles denote age-matched controls (AMC). Crosses denote means, while horizontal lines denote the 25th, 50th and 75th percentiles.

Figs). Trabecular thickness increased with age and was reduced by breeding, but there was no significant interaction between age and breeding (Table 4; S10 Fig).

Cortical area fraction showed significant effects of breeding (lower in bred mice), age (decrease with age), genotype (higher in *Pappa2^{KO/KO}* mice), and a significant interaction between breeding and age, whereby the difference between bred and AMC after 3 weeks of recovery was smaller at 2 months than at 5 or 7 months (Table 4; Fig 4). Cortical thickness and porosity showed similar patterns (Table 4; S11 and S12 Figs). Trabecular bone volume fraction and trabecular thickness showed significant genotype by age interactions, whereby these traits

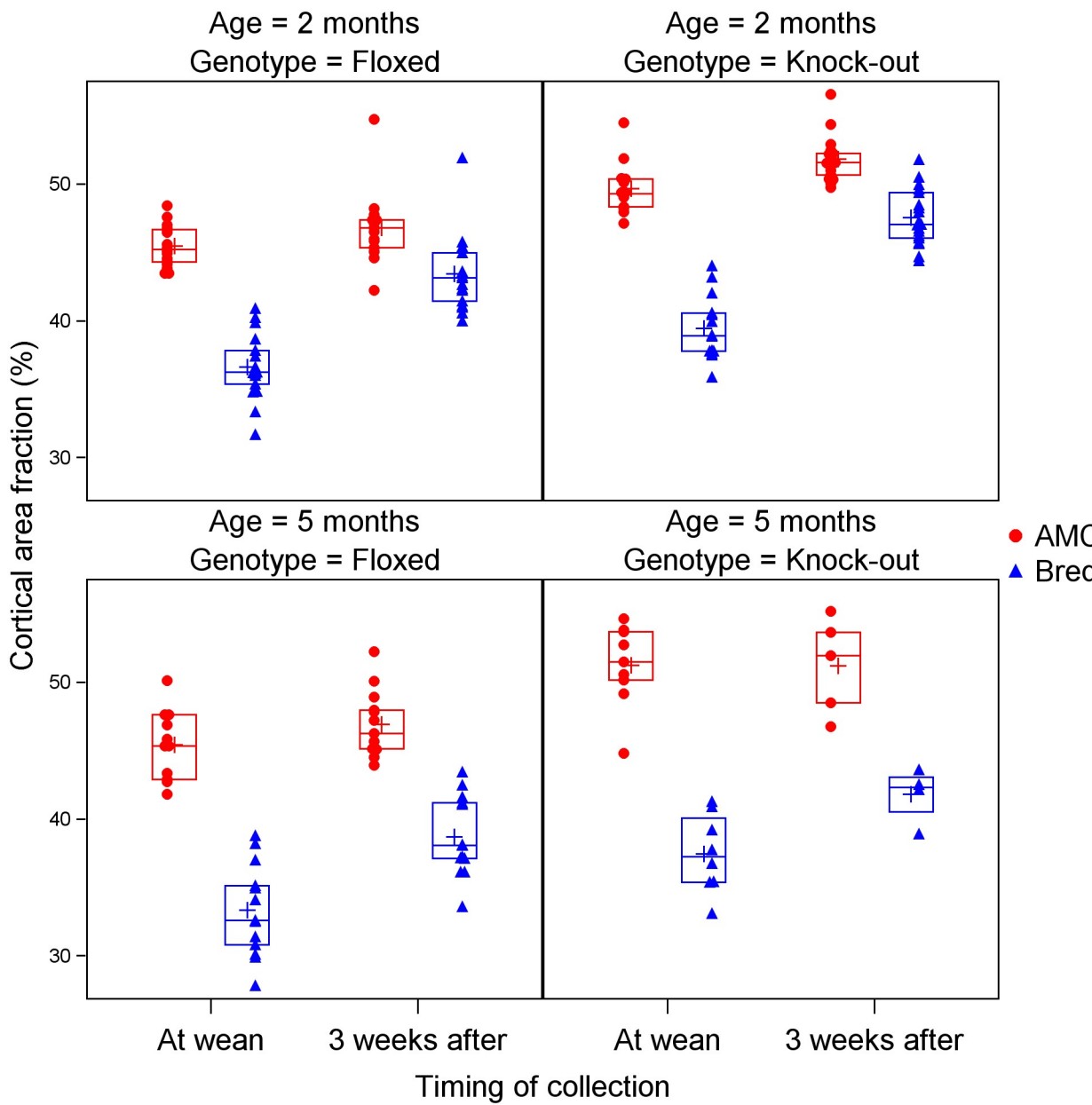

**Fig 2. Effects of age, *Pappa2* genotype, lactation and recovery after weaning on cortical area fraction.** Blue triangles denote bred mice, and red circles denote age-matched controls (AMC). Crosses denote means, while horizontal lines denote the 25th, 50th and 75th percentiles.

were elevated in *Pappa2*$^{KO/KO}$ mice at 5 months but reduced at 7 months (S13 and S14 Figs). These results including 7 month old mice are consistent with the analyses above including only 2 and 5 month old mice collected at both wean and after 3 weeks of recovery: the recovery of cortical bone is impaired at older ages, whereas trabecular bone is not affected by breeding even at wean in older mice, potentially because levels are so low that it cannot provide a substantial amount of calcium.

**Table 4. Effects of age and *Pappa2* genotype on the recovery of skeletal traits in 2, 5 and 7 month old mice, including only mice collected 3 weeks after weaning.**

| | Breeding (bred vs. AMC) | Age (2 vs. 5 vs. 7 months) | Genotype (*Pappa2*$^{KO/KO}$ vs. *Pappa2*$^{fl/fl}$) | Breeding*age interaction | Genotype*age interaction |
|---|---|---|---|---|---|
| **Trabecular** | | | | | |
| Bone volume fraction (%) | 0.54 | <0.0001 | 0.36 | 0.002 | 0.0007 |
| Trabecular number (mm$^{-1}$) | 0.66 | <0.0001 | 0.82 | 0.71 | 0.57 |
| Trabecular separation (μm) | 0.62 | <0.0001 | 0.25 | 0.39 | 0.15 |
| Trabecular thickness (μm) | 0.0002 | <0.0001 | 0.55 | 0.81 | 0.01 |
| **Cortical** | | | | | |
| Cortical area fraction (%) | <0.0001 | <0.0001 | <0.0001 | <0.0001 | 0.11 |
| Cortical thickness (μm) | <0.0001 | 0.02 | 0.01 | <0.0001 | 0.06 |
| Cortical porosity (%) | <0.0001 | 0.01 | 0.02 | 0.0004 | 0.40 |

P-values are from general linear models included effects of breeding, age, genotype, the interaction between breeding and age, and the interaction between genotype and age. AMC = age-matched controls.

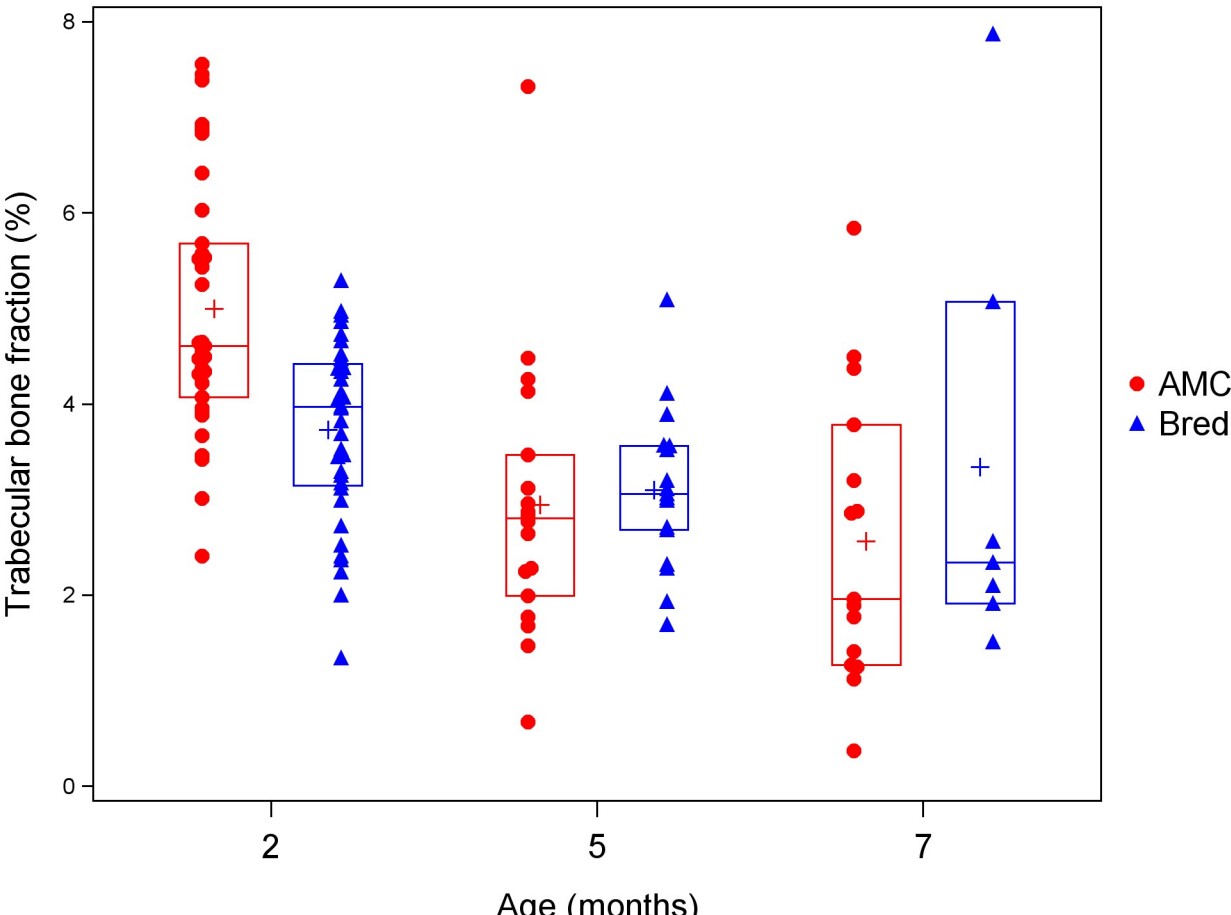

**Fig 3. Effects of age and lactation on trabecular bone fraction among mice collected 3 weeks after weaning.** Blue triangles denote bred mice, and red circles denote age-matched controls (AMC). Crosses denote means, while horizontal lines denote the 25th, 50th and 75th percentiles.

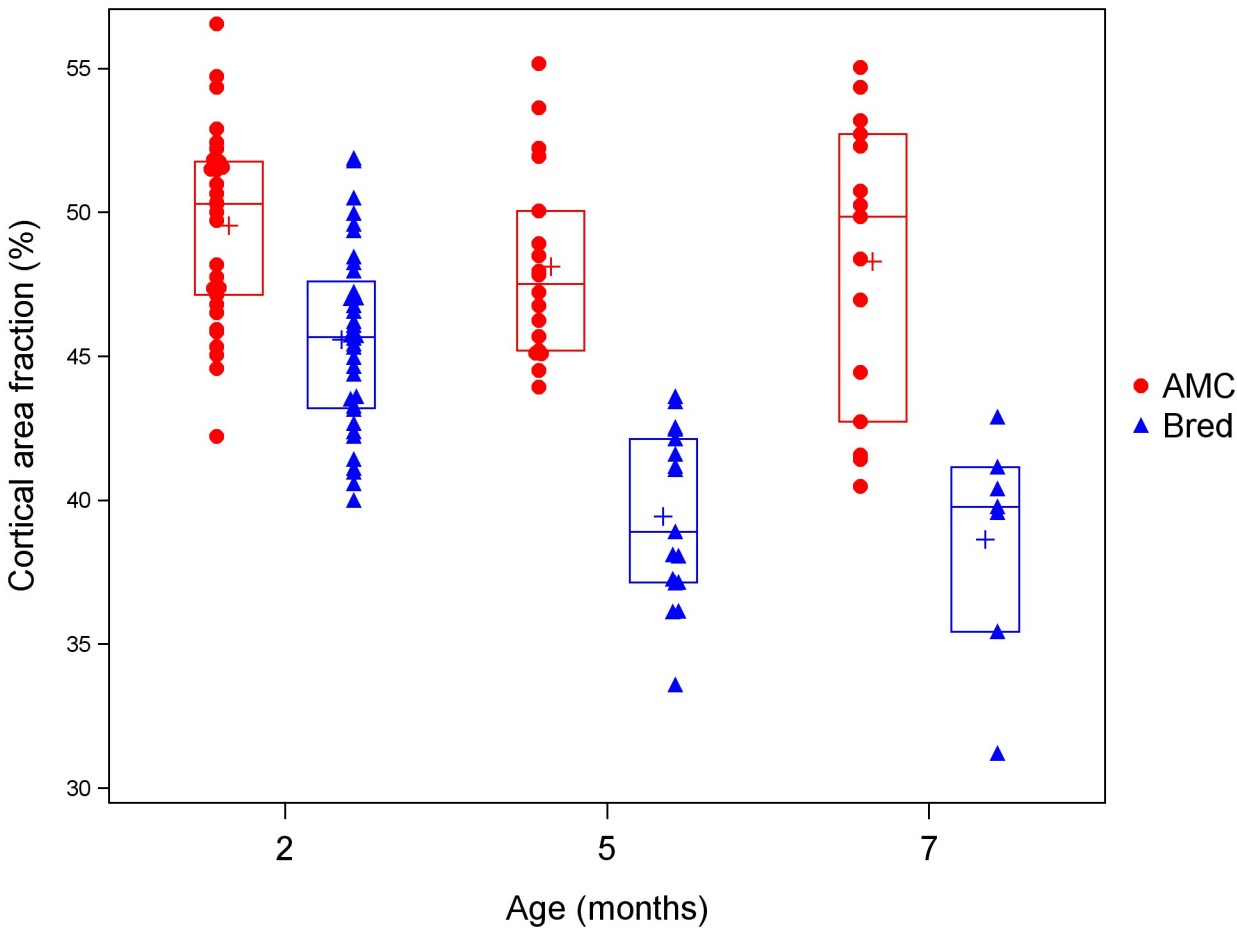

**Fig 4. Effects of age and lactation on cortical area fraction among mice collected 3 weeks after weaning.** Blue triangles denote bred mice, and red circles denote age-matched controls (AMC). Crosses denote means, while horizontal lines denote the 25th, 50th and 75th percentiles.

### Effects of pregnancy vs. lactation on bone

Pregnancy itself affects bone mass and microarchitecture [2, 57], and therefore the effects of lactation described above could have been due, at least in part, to pregnancy. To distinguish between effects of pregnancy and lactation, we analysed females that had given birth, but lost the litter soon after birth, and were collected 3 weeks after birth, i.e., at the time when lactating females would have been collected at wean. We identified 6 such females that could be matched to both a bred female collected at wean and an age-matched control of the same age and genotype collected around the same time. Cortical traits were all significantly reduced in females that had successfully reared a litter, while females that had been pregnant but lost their litters were not significant different from age-matched controls (Table 5). Thus, at 3 weeks after birth, cortical traits were affected by lactation but not by pregnancy. Trabecular thickness showed a similar pattern, although bred females were not statistically different from those who lost a litter (Table 5). Trabecular separation was significantly higher in females that had been pregnant but lost their litters than in age-matched controls (Table 5). While there were no significant differences among groups for bone volume fraction and trabecular number, these traits showed a similar pattern to trabecular separation, i.e., females that had been pregnant

**Table 5. Effects of pregnancy vs. lactation on skeletal traits.**

| | AMC (no pregnancy, no lactation) | Lost litter (pregnancy but no lactation) | Weaned pups (pregnancy and lactation) | Group (AMC vs. lost litter vs. weaned pups) | Cohort |
|---|---|---|---|---|---|
| | | | | P | P |
| Trabecular | | | | | |
| Bone volume fraction (%) | 5.3±0.4[a] | 3.6±0.4[a] | 4.2±0.4[a] | 0.06 | 0.04 |
| Trabecular number (mm$^{-1}$) | 3.2±0.1[a] | 2.9±0.1[a] | 3.1±0.1[a] | 0.08 | 0.0003 |
| Trabecular separation (μm) | 313±11[a] | 360±11[b] | 337±11[ab] | 0.045 | 0.0002 |
| Trabecular thickness (μm) | 44±1[a] | 41±1[ab] | 38±1[b] | 0.02 | 0.005 |
| Cortical | | | | | |
| Cortical area fraction (%) | 48±1[a] | 49±1[a] | 36±1[b] | <0.0001 | 0.014 |
| Cortical thickness (μm) | 193±6[a] | 198±6[a] | 141±6[b] | <0.0001 | 0.23 |
| Cortical porosity (%) | 5.3±0.3[a] | 5.1±0.3[a] | 6.9±0.3[b] | 0.004 | 0.70 |

Six females that lost their litter soon after birth were collected 3 weeks after birth and were matched with a female who bred successfully and an age-matched control (AMC) of the same age and genotype. A matched trio of 3 mice (lost litter, bred successfully and AMC) were considered a cohort. Values are least squares means ± standard error from a general linear model including effects of group and cohort. Values with the same superscript letter are not significantly different.

but lost their litters had the lowest amount of bone (Table 5). These results suggest that these females may have not been able to lactate successfully because of reduced trabecular calcium stores.

## Conclusions

In young mice with intact *Pappa2*, lactation affects femoral trabecular and cortical bone, but only cortical bone shows some recovery 3 weeks after lactation. The deletion of *Pappa2* does not impair this recovery. In mice bred at 5 and 7 months, trabecular bone is no longer affected by lactation, perhaps because levels are so low that it cannot provide a substantial amount of calcium. However, the recovery of cortical bone is impaired at 5 and 7 months. Our results may be relevant to the long-term effects of breastfeeding on the maternal skeleton in humans, particularly given the increasing median maternal age at childbearing [20].

## Supporting information

**S1 Fig. Effects of *Pappa2* genotype, lactation and recovery after weaning on serum IGF-I levels in 2 month old mice.** Blue triangles denote bred mice, and red circles denote age-matched controls (AMC). Crosses denote means, while horizontal lines denote the 25th, 50th and 75th percentiles.
(TIF)

**S2 Fig. Effects of *Pappa2* genotype, lactation and recovery after weaning on serum IGFBP-5 levels in 2 month old mice.** Blue triangles denote bred mice, and red circles denote age-matched controls (AMC). Crosses denote means, while horizontal lines denote the 25th, 50th and 75th percentiles.
(TIF)

**S3 Fig. Effects of age, *Pappa2* genotype, lactation and recovery after weaning on trabecular number.** Blue triangles denote bred mice, and red circles denote age-matched controls (AMC). Crosses denote means, while horizontal lines denote the 25th, 50th and 75th percentiles.
(TIF)

**S4 Fig. Effects of age, *Pappa2* genotype, lactation and recovery after weaning on trabecular spacing.** Blue triangles denote bred mice, and red circles denote age-matched controls (AMC). Crosses denote means, while horizontal lines denote the 25th, 50th and 75th percentiles.
(TIF)

**S5 Fig. Effects of age, *Pappa2* genotype, lactation and recovery after weaning on trabecular thickness.** Blue triangles denote bred mice, and red circles denote age-matched controls (AMC). Crosses denote means, while horizontal lines denote the 25th, 50th and 75th percentiles.
(TIF)

**S6 Fig. Effects of age, *Pappa2* genotype, lactation and recovery after weaning on cortical thickness.** Blue triangles denote bred mice, and red circles denote age-matched controls (AMC). Crosses denote means, while horizontal lines denote the 25th, 50th and 75th percentiles.
(TIF)

**S7 Fig. Effects of age, *Pappa2* genotype, lactation and recovery after weaning on cortical porosity.** Blue triangles denote bred mice, and red circles denote age-matched controls (AMC). Crosses denote means, while horizontal lines denote the 25th, 50th and 75th percentiles.
(TIF)

**S8 Fig. Effects of age and lactation on trabecular number among mice collected 3 weeks after weaning.** Blue triangles denote bred mice, and red circles denote age-matched controls (AMC). Crosses denote means, while horizontal lines denote the 25th, 50th and 75th percentiles.
(TIF)

**S9 Fig. Effects of age and lactation on trabecular spacing among mice collected 3 weeks after weaning.** Blue triangles denote bred mice, and red circles denote age-matched controls (AMC). Crosses denote means, while horizontal lines denote the 25th, 50th and 75th percentiles.
(TIF)

**S10 Fig. Effects of age and lactation on trabecular thickness among mice collected 3 weeks after weaning.** Blue triangles denote bred mice, and red circles denote age-matched controls (AMC). Crosses denote means, while horizontal lines denote the 25th, 50th and 75th percentiles.
(TIF)

**S11 Fig. Effects of age and lactation on cortical thickness among mice collected 3 weeks after weaning.** Blue triangles denote bred mice, and red circles denote age-matched controls (AMC). Crosses denote means, while horizontal lines denote the 25th, 50th and 75th percentiles.
(TIF)

**S12 Fig. Effects of age and lactation on cortical porosity among mice collected 3 weeks after weaning.** Blue triangles denote bred mice, and red circles denote age-matched controls (AMC). Crosses denote means, while horizontal lines denote the 25th, 50th and 75th percentiles.
(TIF)

**S13 Fig. Effects of age and genotype on trabecular bone fraction among mice collected 3 weeks after weaning.** Blue triangles denote *Pappa2*$^{fl/fl}$ mice and red circles denote *Pappa2*$^{KO/KO}$ mice. Crosses denote means, while horizontal lines denote the 25th, 50th and 75th percentiles.
(TIF)

**S14 Fig. Effects of age and genotype on trabecular thickness among mice collected 3 weeks after weaning.** Blue triangles denote *Pappa2*$^{fl/fl}$ mice and red circles denote *Pappa2*$^{KO/KO}$ mice. Crosses denote means, while horizontal lines denote the 25th, 50th and 75th percentiles.
(TIF)

**S1 File. Raw data.**
(XLSX)

## Acknowledgments

We thank Lauren Pettifer, Amritdeep Randhawa and Kaytlyn Tasalloti for assistance with the isolation of bones, and the Animal Care staff at Simon Fraser University for maintaining the animals. We thank Guobin Sun, Jonathan Villareal and Nancy Ford at the Centre for High-Throughput Phenogenomics at the University of British Columbia for performing the micro-CT scans. The Centre for High-Throughput Phenogenomics is supported by the Canada Foundation for Innovation, British Columbia Knowledge Development Foundation, and the UBC Faculty of Dentistry.

## Author Contributions

**Conceptualization:** Julian K. Christians.

**Formal analysis:** Monika D. Rogowska, Uriel N. V. Pena, Julian K. Christians.

**Investigation:** Monika D. Rogowska, Uriel N. V. Pena, Nimrat Binning, Julian K. Christians.

**Writing – original draft:** Julian K. Christians.

**Writing – review & editing:** Monika D. Rogowska, Uriel N. V. Pena, Nimrat Binning.

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
