## [Decision Letter · Decision Letter 0]

4 Aug 2021

PONE-D-21-22335

Recovery of the maternal skeleton after lactation is impaired by advanced maternal age but not by reduced IGF availability in the mouse

PLOS ONE

Dear Dr. Christians,

Thank you for submitting your manuscript to PLOS ONE. After careful consideration, we feel that it has merit but does not fully meet PLOS ONE’s publication criteria as it currently stands. Therefore, we invite you to submit a revised version of the manuscript that addresses the points raised during the review process. 

We look forward to receiving your revised manuscript.

Kind regards,

Subburaman Mohan

Academic Editor

PLOS ONE

Journal Requirements:

2. Please include further information regarding your in vivo study, per our guidelines (http://journals.plos.org/plosone/s/submission-guidelines#loc-animal-research). Specifically, please provide details regarding:

- Source of animals and Description of animal care – Food, water, housing

- Animal health monitoring, including frequency and criteria and any efforts made to reduce suffering and distress, such as administering analgesics

- whether humane endpoints were in place during the study and how they were applied

- the method of euthanasia for the mice

-  any mortality that occurred outside of planned euthanasia or humane endpoints

In addition, as part of your revision, please complete and submit a copy of the Full ARRIVE 2.0 Guidelines checklist, a document that aims to improve experimental reporting and reproducibility of animal studies for purposes of post-publication data analysis and reproducibility: https://arriveguidelines.org/sites/arrive/files/Author%20Checklist%20-%20Full.pdf (PDF). Please include your completed checklist as a Supporting Information file. Note that if your paper is accepted for publication, this checklist will be published as part of your article.

Reviewers' comments:

Reviewer's Responses to Questions

**Comments to the Author**

1. Is the manuscript technically sound, and do the data support the conclusions?

Reviewer #1: Yes

Reviewer #2: Yes

2. Has the statistical analysis been performed appropriately and rigorously? 

Reviewer #1: Yes

Reviewer #2: Yes

3. Have the authors made all data underlying the findings in their manuscript fully available?

Reviewer #1: Yes

Reviewer #2: Yes

4. Is the manuscript presented in an intelligible fashion and written in standard English?

Reviewer #1: Yes

Reviewer #2: Yes

5. Review Comments to the Author

Reviewer #1: Authors have examined the effects of lactation on femoral trabecular and cortical bone. They evaluated the effects of breeding, timing (pre- and postweaning), aging (2, 5 and 7 months) and genotype (Pappako/ko vs. Pappafl/fl). Results clearly indicated a reduction of bone traits by lactation, but only cortical bone was recovered in 2 months aged mice. Pappa2 deletion, and the disturbance of IGF1 and IGFBP5, did not enhanced the reduction or blocked the recovery. However, aging (5 and 7 months) affects cortical bone recovery in bred mice.

A very interesting study that shows the role of lactation and after lactation (postweaning) on bone traits during aging. I have two questions about the rationale of the main results:

1. If I can understand, authors did not find impairments in cortical bone recovery of Pappa2 deletion mice at 2, 5 and 7 months old (genotype effect). But, I would like to know if authors found genotype*age interaction and single effects of Pappa2 deletion on bone traits at 2, 5 and 7 months in mice collected 3 weeks after weaning compared to the respective Pappa2 floxed control mice and AMC mice. Table 4 did not describe this important question.

2. The effects of breeding and recovery on bone traits were not associated with IGF1 impairments. However, could IGF1 impairment in Pappa2 ko mice be associated with changes in bone traits such as trabecular thickness and cortical area fraction?

Reviewer #2: The main goal of this study was to understand how age of pregnancy, lactation, and IGF-1 state affect bone. The authors used control and PAPPA2 KO dams at 2, 5, and 7 months of age and evaluated their femurs 3 weeks postpartum either with or without lactation period. Age matched virgin females were used as controls. The study included sufficient sample size, it was well designed and controlled for litter size of the lactating dams. Cortical and trabecular compartments of the femur were analyzed by mCT. Serum ELISA assays were used to determine the levels of IGF-1 and the IGFBP5.

The authors found that 2 months old dams recovered their cortical bone traits after lactation, independent of PAPPA2 state. Older dams (at 5 and 7 months old) were not able to recover cortical bone parameters. No changes were observed in trabecular bone of older dams, likely due to the low basal BV/TV% seen in that age.

Overall, the study is informative and has interesting implications to late pregnancies and breastfeeding in humans.

However, there are several concerns that need to be addressed:

1. The study lacks mechanistical outcome.

2. Justification of the skeletal site chosen for analysis; Loss of trabecular bone during pregnancy and lactation is often seen in the vertebral body. The trabecular bone at the femur distal metaphysis reduces dramatically between 2-4 months even in virgin mice.

3. Were calcium levels measured in serum/urine during pregnancy/lactation?

4. What is the predicted mechanism by which cortical bone area reduces after pregnancy/lactation? Endocortical resorption? Any evidence?

5. What is the predicted mechanism by which cortical bone area recovers? Periosteal/endosteal apposition?

6. Was the food intake and metabolic rate of the PAPPA2KO dams similar to that of controls?

6. PLOS authors have the option to publish the peer review history of their article (what does this mean?). If published, this will include your full peer review and any attached files.

Reviewer #1: No

Reviewer #2: No

---

## [Author Response · Author response to Decision Letter 0]

13 Aug 2021

Our letter responding to each point has been uploaded.

---

## [Editor Report · Decision Letter 1]

18 Aug 2021

Recovery of the maternal skeleton after lactation is impaired by advanced maternal age but not by reduced IGF availability in the mouse

PONE-D-21-22335R1

Dear Dr. Christians,

We’re pleased to inform you that your manuscript has been judged scientifically suitable for publication and will be formally accepted for publication once it meets all outstanding technical requirements.

Kind regards,

Subburaman Mohan

Academic Editor

PLOS ONE
---

## [Editor Report · Acceptance letter]

23 Aug 2021

PONE-D-21-22335R1 

Recovery of the maternal skeleton after lactation is impaired by advanced maternal age but not by reduced IGF availability in the mouse 

Dear Dr. Christians:

I'm pleased to inform you that your manuscript has been deemed suitable for publication in PLOS ONE. Congratulations! Your manuscript is now with our production department. 

Kind regards, 

on behalf of

Dr. Subburaman Mohan 

Academic Editor

PLOS ONE